# Is Organic Food Becoming Less Safe? A Longitudinal Analysis of Conventional and Organic Product Recalls

Mesbahuddin Chowdhury [1,*] , Pavel Castka [1,*] , Daniel Prajogo [2], Xiaoli Zhao [3] and Lincoln C. Wood [4,5]

1   Department of Management, Marketing and Entrepreneurship, UC Business School,
    University of Canterbury, Christchurch 8140, New Zealand
2   Department of Management, Monash Business School, Monash University,
    Caulfield East, VIC 3145, Australia; daniel.prajogo@monash.edu
3   Cognitia International, Christchurch 8025, New Zealand; lillianzhaoxl@gmail.com
4   Department of Management, Otago University, Dunedin 9054, New Zealand; lincoln.wood@otago.ac.nz
5   School of Management, Curtin University, Perth, WA 6102, Australia
*   Correspondence: mesbahuddin.chowdhury@canterbury.ac.nz (M.C.); pavel.castka@canterbury.ac.nz (P.C.);
    Tel.: +64-3369-3711 (M.C.)

**Abstract:** Organic products are often portrayed as a healthy alternative—grown in a sustainable way, often locally and subject to external certification scrutiny. However, recent high-profile cases of contaminated organic food have raised questions about the risks associated with organic produce: is organic produce becoming less safe and more risky? The context for this investigation is in the realm of food product recalls. Based on 2010–2017 panel data from the US on food product recalls (with 2721 observations), this paper compares the volume of recalls (adjusted for the growth of sales) between conventional and organic food. This paper further addresses two food-related risks: design risk (a risk that is present in the development of food; such as the use of unapproved ingredients or the omission of some ingredients on the food label) and process risk (a risk within the supply chain, such as the contamination of food products with salmonella or *E. coli*). Further comparison is drawn based on food product type (here the paper distinguishes between processed and unprocessed food). The paper demonstrates that organic products are becoming less safe and that organic products are recalled at a higher rate. In comparison to conventional produce, organic produce is more prone to process risk and far less to design risk. Similar conclusions are reached even when the organic produce is analysed from a product type perspective.

**Keywords:** conventional food; organic food; supply chains; risk; product recalls



## 1. Introduction

Organic food has grown in popularity and its market share, as well as consumer acceptance, have rapidly grown across the globe [1]. According to IFOAM [2], by the end of 2017, at a global level, organic agriculture was practiced in 181 countries, over a total of approximately 69.8 million hectares (1.4% of agricultural land), and the size of the organic market reached USD 97 billion. In general, organic production systems adhere to specific requirements [3] such as a reduced use of chemicals (inclusive of synthetic pesticides, fertilizers, antibiotics, growth hormones) whilst also adhering to ethical practices (e.g., animal welfare) and social responsibility (e.g., workers' rights and ethical trade). In comparison to conventionally produced food, organic food is expected to be more nutritional and safer to consume [4].

Numerous studies have provided analyses of the differences between organic and conventionally produced food—in terms of nutritional value [1,5], the drivers influencing farmers' decision to adopt organic practices [6], consumer perception of the benefits [7,8], and their impact on health [5,9]. The evidence about the differences between organic and conventionally grown food is often inconclusive, and there is a need to scrutinize these

differences further [10]. Bourn and Prescott [11] single out, in particular, the question of microbiological risk and argue that "the question of whether the consumption of organically grown food confers any greater microbiological risk to consumers than conventional food has not yet been addressed in a scientific manner (Bourn and Prescott, 2002, p. 24)". The differences between organic and conventional foods have been compared in the literature from numerous angles. For instance, one literature review on such comparison lists factors such as "economics, crop yields, agronomic factors (soil chemical properties, soil physical properties, soil microbiological activity, pest and disease burdens etc.), farm management practices, product quality (nutritional value, taste, shelf life), environmental impacts, biodiversity, farm nutrient inputs and social, trade, and political issues associated with food production" [11]. The literature reviews in this area of study are generally organized around issues related to health, nutritional and safety characteristics; other authors add ethical, environmental and social issues related to organic production [10,11]. In this paper, we extend this scope by considering the risk associated with contamination in food production systems. Specifically, we compare conventional and organic food systems.

Food production systems are amongst the most complex supply chains that encompass farming, food processing, distribution and retail [12]. The entire "farm-to-plate" supply chain is susceptible to multiple process-based risks: microbiological contamination (in the entire supply chain), particle contamination (typically during food processing), health risks due to undeclared ingredients or the use of unapproved additives [13]. In comparison to conventional food production systems, organic food production systems are even more complex because of the extra constraints dictated by organic food rules—such as the limited use of chemicals [5].

Recent media coverage has linked organic produce to an increased health risk and scrutinized various high-profile cases of organic produce contamination. For instance, a 2011 *E. coli* outbreak linked to an organic farm killed 22 people and made more than 2200 sick. This case was widely covered across the globe [14], which raised concerns about organic food safety in general. In 2015, *Wall Street Journal* reported that "organic food is less safe" [15]—pointing out the increased product recalls of organic food. Even though media coverage does not necessarily advise consumers to shy away from organic produce, it is clear that organic food systems face challenges [16]—especially because of the steadily growing demand for organic produce and anecdotal evidence about the risks in the organic food production systems.

This paper aims to address the need to investigate the food safety of organic produce. The overarching research for our study is whether organic food is becoming less safe. We studied this question in the context of food product recalls and uses the 2010–2017 panel data from the US on food product recalls (a total of 1892 food-related recalls were used in our analysis). In guiding our investigation, we developed four research questions, and we started by examining the proportion of the product recalls on organic food produce over time which reflect the trend of the risks inherent in those products. Therefore, we ask: *Research Question 1: Does the proportion of organic product recalls increase over time?*

Secondly, due to the strong growth trend in consumer demand for organic produce, we anticipate many producers switching to organic food production systems or rapidly expanding their organic systems. As this rapid shift occurs, there are increasing opportunities for risks and errors to occur, particularly because of the additional complexity of organic food production [13] such as constraints on the use of chemicals and applications. Therefore, we aim to examine whether the risks of errors occurring in organic products is greater than that in non-organic (conventional products) as reflected in the volume of recalls (adjusted for the growth of sales) between conventional and organic food. This point is important since greater risks of errors among organic products might offset the potential benefits they offer. As such, we ask: *Research Question 2: Do organic products become more prone to errors compared to conventional products?*

Third, our study aimed to conduct further analysis to locate the occurrence of the risks. In this regard, we focused on two aspects of food-related risks: design risk (a risk

that is present in the development of food such as the use of unapproved ingredients or the omission of some ingredients at the food label) and process risk (a risk within the supply chain, such as the contamination of food products with salmonella or *E. coli*). Even though the recalls of conventional and organic produce might occur for similar reasons (e.g., incorrect storage; incorrect content disclosure), it is unclear to what extent the recalls due to design and process reasons might differ between conventional and organic food. Therefore, we ask the following questions: *Research Question 3: Is there any difference in the causes of product recalls between organic and non-organic products?*

Fourth, the paper also compares two distinctive food product types—processed and unprocessed food. Given the differences in complexities and constraints between organic and non-organic food systems and whether the foods are processed or unprocessed, there are likely to be different causes of recalls. *Research Question 4: Are there any differences in the causes of product recalls between processed and unprocessed food among conventional and organic products?*

This research is important due to the increased consumer interest in organic food consumption and the scaling up trends in organic food production systems. Consequently, organic produce might be susceptible to high risk in terms of food safety, as suggested by WSJ [15], however, such a question has been neglected in the academic literature. Specifically, this paper contributes to the growing number of comparative studies on conventional and organic food and provides a comparison of safety in conventional and organic food production systems. More specifically, it further investigates the contamination of organic and conventional produce based on the nature of their production. In terms of practical implications, this paper provides an insight for consumers in terms of food safety and for food companies, it unravels the potential risks in supply chains that may damage to firm reputation [17,18] and induce an economic cost to firms from product recalls over a range of industry sectors [17,19], including food [20,21].

This article is structured as follows. We first review the characteristics of organic and conventional food production systems, explaining the background for the four research questions. We then explain the dataset used in the study and our analytical approach. We then present the results, explain the importance of the results and highlight questions for the future research. This paper concludes with guidance for consumers, managers and policymakers.

## 2. Literature Review

### 2.1. Organic Food—Definition and Background

The International Federation of Organic Movements (2018) defines organic agriculture as "a production system that sustains the health of soils, ecosystems and people; relies on ecological processes, biodiversity and cycles adapted to local conditions, rather than the use of inputs with adverse effects; and combines tradition, innovation and science to benefit the shared environment and promote fair relationships and a good quality of life for all involved." The requirements for organic produce differ globally. Though the schemes have some differences, the requirements generally forbid "synthetic pesticides or fertilizers or routine use of antibiotics or growth hormones" [10].

The demand for organic food has been steadily growing. By the end of 2017, the organic food market was valued at EUR 40 billion in the US [2] and EUR 37.3 billion in Europe [22]. The rise in demand for organic produce is linked to an increased preference for organic food from consumers. On the one hand, the demand is driven by increased awareness about the environmental and social responsibility of organizations. Consumers also have inflated beliefs in food and often perceive organic products to be more nutritional and less contaminated—despite less convincing scientific evidence [23,24]. On the other hand, this rise in demand is also driven by retailers, who turn to suppliers of organic and socially responsible produce. This trend is not only driven by consumers but also by investors and retailers [25].

## 2.2. Food Production Systems

A food production system is generally described as a farm-to-fork value chain. The main stages in the food production system include farm management, food processing, distribution and retail [12]. In farm management, the key elements are farming inputs—e.g., seeds, feed, fertilizers or pesticides; firm resources—e.g., farmland parcels, stables and machinery; and agricultural products—e.g., cattle and produce. These various elements contribute to the production of agri-products that are either further processed into final food products (i.e., a can of fish) or packaged, such as fresh products that are directly packed without processing. In the last stage of the food production system, products are shipped in different containers, distributed to retailers and sold to consumers [12,26].

The food production system is relatively complex in nature. It significantly differs between the types of production: e.g., livestock farming, arable farming and greenhouse cultivation. A common feature of agricultural production is that it depends on natural conditions such as climate (day length and temperature), soil, pests, diseases and weather. Food processing widely varies as different food products can be produced by adopting different processing techniques [27]. It was also characterized by a combination of continuous or batch processing and discrete processes after packaging. In addition to that, there are many diverging and converging processes and by-products available which combine different objects into a single object (e.g., blending) or split into multiple objects (e.g., slaughtering). The distribution of food products combines high volume with frequent delivery and increasingly intricate distribution [28]. Processes can also vary depending on the distribution network layout, including different consolidation strategies and different modes of transportation. Food retailing processes are diverse due to different outlet channels, for example supermarkets, specialized food shops, food service provider including restaurants and caterers as well as increasingly popular webshops [12].

Food production systems are subject to high levels of scrutiny due to the level of risk to consumers. Regulations, policies and standards for food safety have been developed for the food industry over time [29]. A range of inspection, testing and conformity assessments exist across the food system [30] and firms hire a certifier/auditor (or are subject to governmental inspections) to ensure/validate that they have met the certification standards [31]. For instance, food firms around the world are increasingly using standardized quality assurance systems to improve the quality and safety of food products, production and supply chain processes and seek external audit and certification [32–34]. The three most important generic quality assurance systems in the food sector are Good Agricultural Practices (GAPs), Hazard Analysis of Critical Control Points (HACCPs) and international standards by International Organization for Standardization (ISO) such as ISO 22000 [35,36]. Governments also play an essential role in providing policy guidance on the most appropriate quality assurance systems and verifying/auditing their implementation as a means of regulatory compliance [35].

Organic production systems differ from conventional food systems in several ways. First, organic production systems have to adhere to the very specific requirements of organic production. In farm management [37], it means, for instance, that the farmers cannot use pesticides (plant farms) or antibiotics (animal farms). Organic farms also have to adhere to other specific practices, such as crop rotation and cover cropping [38]. Such requirements mean that organic producers need to modify their operations, and apart from increased cost, the producers face several challenges [39]. For instance, the reduced (or forbidden) use of pesticides affects the effectiveness of weed management; the produce may be more susceptible to pest insects and plant pathogens [38] and the labour cost is higher [40]. At the food processing stage, the manufacturers might have to use dedicated equipment for organic produce and have to source from specialized suppliers. Organic food processing is also governed by a set of principles, such as naturalness and focus on minimal, sustainable and careful processing [41].

### 2.3. Safety Risks in Food Systems

Food safety can be jeopardized at any stage of food systems or product life cycle. A simple categorization of stages where food safety can be affected includes the design phase and the process phase. During the design of a product, errors and mistakes can be made through the inclusion of unapproved food ingredients, the inadequate design of food processing and packaging or poor (or misleading) instructions for use. Product developers can also fail to disclose allergens or the risk of cross-contamination due to the use of the same equipment across multiple product lines [42]. These issues can be managed through effective product quality design processes [43]. On the other hand, process errors can occur at any stage of a product's life cycle—in farming, processing, or handling (which is often the case in a co-operative processing facility or the use of contaminated water from a local supply) in the manufacturing processes or in transport. Third-party logistics providers often provide several movements of the items and temperature fluctuations can affect the safety of the processed food products. Food products can suffer from microbiological contamination, the contamination of raw materials, poor sanitation, or from the presence of foreign objects. [44]. The consequences of safety problems with food include costs to individuals (e.g., pain, suffering, and medical costs); industries or companies (e.g., recall costs and plant clean-ups), and the public health costs such as clean-up costs and administrative costs related to investigations [20].

Despite the plethora of preventative measures in food systems (monitoring of hazard points, inspections, product testing, and the certification of food management systems), sometimes errors can be prevented and food that poses a health risk (i.e., contaminated food) is distributed in supply chains. In such cases, the affected food is recalled.

Food recalls represent an important mechanism in food systems management. Recalls can be voluntary (based on a firms' discretion) or involuntary (required by a government agency, such as the Food and Drug Administration (FDA) in the US). Increasingly, firms often take a more voluntary and proactive stance, but there is mixed evidence on the corporate benefits of this proactivity (Zhao et al., 2013). Not all risks and the associated recalls are treated equally; most jurisdictions impose a gradated scale. For example, the FDA [45] uses the following classification:

- Class I recall: severe, suggesting reasonable probability of lasting adverse health consequences or death. Examples include alfalfa sprouts contaminated with Salmonella spp; under-processed chilli containing Clostridium botulinum toxin; and products containing undeclared allergens [46];
- Class II recall: may cause temporary/reversible harm but a remote probability of the adverse health consequences;
- Class III recall: least severe, not likely to cause adverse health consequences

The recall system under FDA governance provides a useful example of food recall management that is similar across the world. The FDA is able to mandate a recall when a firm elects to not conduct a voluntary recall, and the FDA determines that there is a reasonable probability of the adulteration of the food product or it has been misbranded and when there is reasonable probability that exposure may cause serious adverse health consequences or death to humans or animals (SAHCODHA) [46]. The FDA will notify the firms and provide the opportunity to voluntarily initiate recalls. If this does not occur in a timely manner or is refused, the FDA may commence a mandatory recall. Risks that the FDA would consider to represent serious adverse health consequences include "Listeria monocytogenes (Lm) or Salmonella spp. in Ready-to-eat foods, certain undeclared allergens in food products, *E. coli* O157:H7 in leafy greens, and botulinum toxin found in food products" [46]. At the firm level, a firm or a manager should notify their board and legal department, employees, appropriate government agencies (e.g., the FDA in the US), distributors or retailers downstream, announce to shareholders, and seek to notify consumers of the product [44].

## 3. Research Method

### 3.1. Data Collection

The dataset was created from two data sources. First, the data on recalls were downloaded from the FDA website (http://www.fda.gov/Safety/Recalls/ArchiveRecalls/default.htm) (accessed during July–August 2019) and transferred into an excel spreadsheet. The dataset contains all food-related recalls between 2010 and 2017. Each recall data point provides information on the producer, product description, reason/problem, details about the recall (recall announcement) and a photo of the packaging (the photo was used to verify whether the product carries any form of formal certification and whether the product is sold as organic). The data from FDA are verbatim (e.g., the reason for recall is described by FDA in brief verbatim statements such as "incorrect concentration listed"; the product description as "Rock Hard Extreme and Passion Coffee Dietary Supplements"). Second, the data on food sales and organic food sales were downloaded from the Organic Trade Association's 2018 Organic Industry Survey. The survey was conducted between 25 January 2018 and 26 March 2018. The sales were used to adjust the data over time. We downloaded the data for food sales and organic food sales during the period 2010–2017 to match our product recall data. The results are presented in Table 1.

**Table 1.** Sales trends for food and organic food sales.

| Year | Food Sales (in USD Million) | Organic Food Sales (in USD Million) | % of Organic Food Sales from the Total Food Sales |
|---|---|---|---|
| 2010 | 677,354 | 22,961 | 3.4% |
| 2011 | 713,985 | 25,148 | 3.5% |
| 2012 | 740,450 | 27,965 | 3.8% |
| 2013 | 760,486 | 31,378 | 4.1% |
| 2014 | 787,575 | 35,099 | 4.5% |
| 2015 | 807,998 | 39,006 | 4.8% |
| 2016 | 812,907 | 42,507 | 5.2% |
| 2017 | 822,160 | 45,209 | 5.5% |

### 3.2. Data Preparation and Coding

In the next step, the verbatim statements were categorized. *Reasons for Recall* were categorized as *Design Reasons* and *Process Reasons*. *Design Reasons* refer to instances where the recall relates to risky issues originating in the design stage of the product lifecycle. For instance, a firm uses an unapproved supplement ("unapproved new drug—contains sibutramine"); the product is recalled due to faulty packaging ("Bottles may break during opening") or due to undeclared ingredients ("Undeclared soy and wheat"). *Process reasons* are classified into *Bacterial Contamination* (for example—a product has been contamination by salmonella, listeria and other) and *Particle Contamination* (for example—represented by reasons such as "Presence of metal fragments" or "May contain shredded plastic fragments").

We also classified the food products in recalls into two categories—*unprocessed* and *processed products*. *Unprocessed product* is defined as a product that is either raw (seeds, nuts, eggs, fresh sandwiches; fresh fruit and vegetables); seasoning mixes and seasoning powders and a product that is not processed (heated) such as ice cream, frozen fish, etc. or partially heated, such as cold-smoked fish or prepared from fresh ingredients (salads, sandwich, hummus and similar). *Processed products* are characterized by heat preservation— such as bakery, canned food as well as products preserved through pasteurization and supplements (vitamins, health and sexual performance). The categorization was coded by the authors, and any discrepancies were resolved through a consensus-based discussion. The vast majority of the coding was very straightforward (as demonstrated by the examples in the text above), and the only discrepancies were due to poorly formulated reasons in the FDA data. In such instances, the team went back to the product recall data point to verify the reason for the recall.

*3.3. Data Analysis*

In this study, we mainly adopted descriptive data analysis approaches (e.g., frequencies, percentages) to address the research questions. Most of the variables used in this study were coded as binary variables (1 = present and 0 = absent). For example, we coded each recalled food product from 2010 to 2017 as organic or conventional (1 = organic, 0 = conventional). For the analysis, we determined the frequencies for these two categories for each year and these values represent how many recalls are associated with organic food products and how many with conventional food products. The frequency of organic food products' recall has been used to address research questions 1 and 2. We coded the cause for each product recall event from 2010 to 2017 using design reasons (if this is the cause, then put 1, otherwise 0) or process reason (if this is the cause, then put 1, otherwise 0). Similarly to a previous procedure, we added all numbers under each of these two broad categories of reasons for each year and these numbers represent the frequencies, i.e., how many times a *Design reason* or *Process reason* is the cause of food product recall in each year. We used these two variables to address our research questions 3 and 4. We also categorised recalled food products into unprocessed and processed. For example, if the recalled food product is under an unprocessed category, we used 1, otherwise 0. Similarly, if the recalled food product is under the process category, we used 1, otherwise 0. We added all these numbers under these two product categories so these numbers represent the frequencies for each year. These variables were used to address research question 4.

## 4. Results

We structured this section by presenting the results of our data analysis following the four research questions we listed at the outset of this study.

*4.1. Research Question 1: Do Organic Products Become More Prone to Errors?*

To address this question, we analysed the growth of organic food product recalls between 2010 and 2017. Taking 2010 data as the baseline (and accounting for the growth of sales of organic food), the paper determines the expected level of recalls and compares the actual recalls against the expected levels. The growth percentage for organic food sales during the period 2011–2017 were calculated using the formula: growth percentage of sales at $year_t$ = (sales at $year_t$—sales at $year_{t-1}$)/sales at $year_{t-1}$. The expected product recall was calculated using the formula: expected product recall at $year_t$ = expected product recall at $year_{t-1}$ + (expected product recall at $year_{t-1}$ × the growth percentage at $year_{t-1}$). However, for estimating the expected organic product recall in 2011, we used the actual product recall in 2010 (that is 9) as the expected product recall at $year_{t-1}$. The results are presented in Table 2.

**Table 2.** The actual and expected growth of sales and recalls for organic food.

| Year | Organic Food Sales (in USD Million) | % Growth of Organic Food Sales | EXPECTED Product Recall for Organic Food | ACTUAL Organic Food Product Recalls |
|---|---|---|---|---|
| 2011 | 25,148 | 9.5% | 10 | 17 |
| 2012 | 27,965 | 11.2% | 11 | 25 |
| 2013 | 31,378 | 12.2% | 12 | 11 |
| 2014 | 35,099 | 11.9% | 14 | 34 |
| 2015 | 39,006 | 11.1% | 15 | 30 |
| 2016 | 42,507 | 9.0% | 17 | 33 |
| 2017 | 45,209 | 6.4% | 18 | 28 |

By comparing columns 3 (expected product recalls for organic food) and 4 (actual product recalls for organic food) in Table 2, we can conclude that the numbers of actual organic food product recalls are consistently higher than their expected numbers based on

the growth rate of the organic food product sales, except in 2013. The results suggest that recalls of organic food products exceed what we might expect from a naive analysis.

### 4.2. Research Question 2: Does the Proportion of Organic Product Recalls Increase over Time?

In addressing this question, we analysed the trend of the proportion of organic product recalls over the eight-year period. The results, as shown in Table 3 (and Figure 1 for the graphical presentation), show an increasing trend. Coupled with the proportion of organic product sales that shows a small increment during that period, this trend strengthens the case of an increased risk associated with organic products.

**Table 3.** Comparison of sales and recalls between conventional and organic products.

| Year | Total Food Product Sales (in USD Million) | Organic Food Product Sales (in USD Million) | % of Organic Food Product Sales | Total Food Product Recalls | Organic Food Product Recalls | % of Organic Food Product Recalls |
|---|---|---|---|---|---|---|
| 2010 | 677,354 | 22,961 | 3.4% | 288 | 9 | 3.13% |
| 2011 | 713,985 | 25,148 | 3.5% | 312 | 17 | 5.50% |
| 2012 | 740,450 | 27,965 | 3.8% | 374 | 25 | 6.95% |
| 2013 | 760,486 | 31,378 | 4.1% | 272 | 11 | 4.07% |
| 2014 | 787,575 | 35,099 | 4.5% | 291 | 34 | 11.68% |
| 2015 | 807,998 | 39,006 | 4.8% | 355 | 30 | 8.47% |
| 2016 | 812,907 | 42,507 | 5.2% | 486 | 33 | 6.79% |
| 2017 | 822,160 | 45,209 | 5.5% | 346 | 28 | 8.09% |

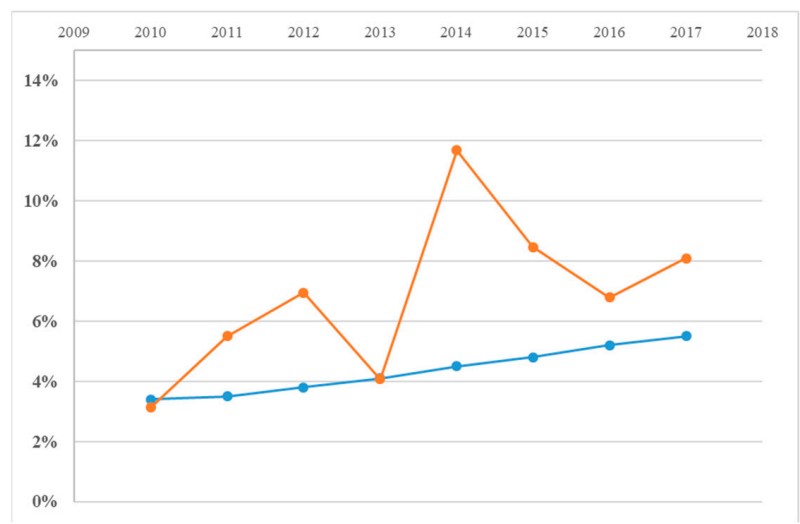

**Figure 1.** Comparison between the growth of organic food product sales and organic food product recalls.

Having shown in the previous research question that the number of product recalls of organic food has an increasing trend, we will now compare the proportion and the trend of product recalls between organic and non-organic food based on the data from the eight-year period in Table 3. In particular, we compare the trend of the proportion of the sales of organic food products from the total sales of food products (column 4) with the trend of the proportion of the recalls of organic food products from the total recalls of food products (column 7). Our observation suggest that both columns show a positive trend (growth), but the growth in column 4 (from approximately 3% to approximately 5%) is relatively smaller compared to the growth of column 7 (from approximately 3% to approximately 8%). We also perform further analysis to compare the trends of the two columns by estimating the slopes of the data from the eight-year period. The results show that the slope of column 4 is 0.0032 while the slope of column 4 is 0.0064—indicating that the growth of column 7 is higher (double) than that of column 4. Since the growth of the

proportion of organic food product recalls is higher than the proportion of organic product sales (which represents the volume), we can not only conclude that there is a growth in product recalls for organic food products, but also that its growth is higher than that of non-organic food products.

### 4.3. Research Question 3: Is There Any Difference between Organic and Non-Organic Product Recalls in Terms of the Causes of the Recalls

To account for the reasons for recalls, this paper uses a ratio of design (D) and process (P) recall reasons (D/P ratio) and compares the ratios between conventional and organic products. The D/P ratio is calculated as a fraction of design and process reasons; where D/P > 1 signifies a prevalence of design reasons, D/P < 1 signifies a prevalence of process reason. The higher (lower) the value, the more significant the prevalence of design (process) reasons. To demonstrate the calculation, for instance, in 2011 (in conventional food systems), there were 152 recalls due to design reasons and 140 due to process reasons; therefore, the D/P ratio = 152/140 = 1.09; this shows a slight prevalence of design reasons for a recall within the conventional food category. Figure 2 compares the recall reasons for conventional and organic produce. From Figure 2, we can see that the ratio of design and process reasons is constantly higher for conventional produce.

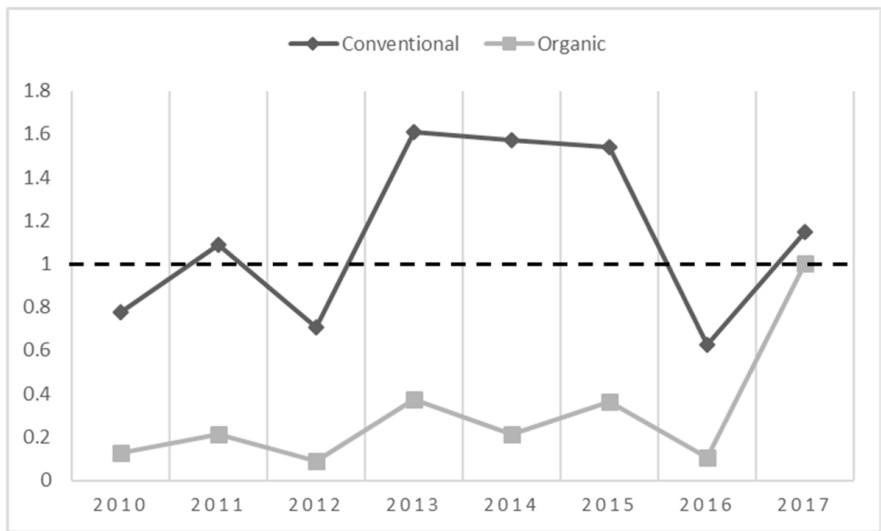

**Figure 2.** Ratio of design and process reasons for recalls.

### 4.4. Research Question 4: Are There Any Differences in the Causes of Product Recalls between Processed and Unprocessed Food among Conventional and Organic Products?

In performing a comparative analysis between unprocessed and processed food products in both conventional and organic production, we used similar D/P ratios (as described above). The results are shown in Figure 3.

The results in Figure 3 show no difference in terms of the causes of product recalls as reflected in the D/P ratio among organic products as both processed and unprocessed organic food products have the proportion of their recalls dominated by process than design (D/P ratio < 1). On the other hand, for conventional food products, the ratio between design and process is split between processed and unprocessed food products with processed food products and unprocessed food product recalls being dominated by process errors (D/P ratio < 1) while processed food product recalls are dominated by design errors (D/P ratio > 1).

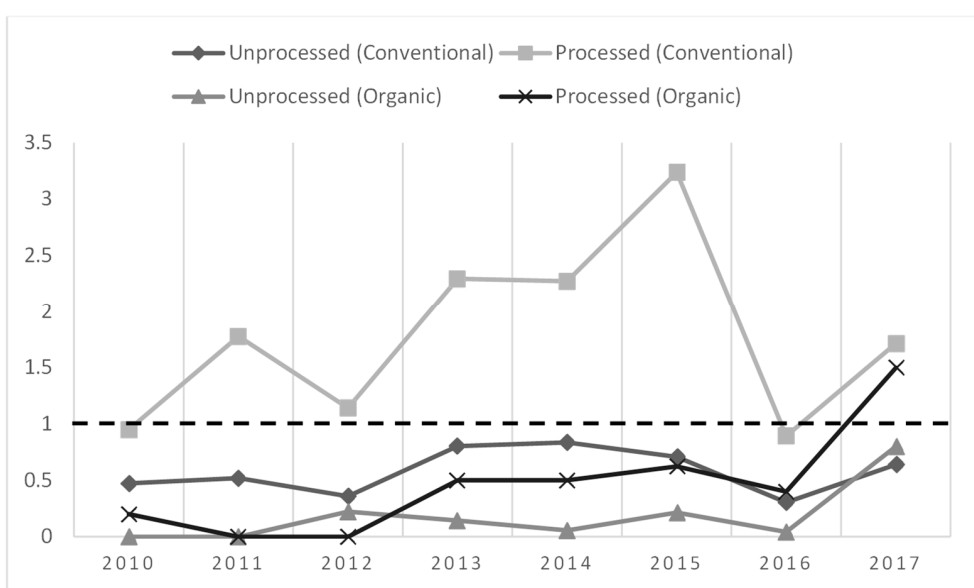

**Figure 3.** Comparison of the ratio of reasons for product recalls between unprocessed and processed food.

## 5. Discussion

The analysis of the data collected during the period 2010–2017 offers three important findings. First, there is an increasing level of product recalls of organic produce, suggesting that the risk associated with organic produce has increased over time. Based on a sample of 2721 observations, of panel data from 2010–2017, drawn from US food product recalls, we found that after adjusting for the annual growth of sales, the growth of product recalls is accelerating. In part, the increasing number of recalls have been attributed in the literature to the improved capability of the detection mechanism in supply chains [42,47]. For instance, improvements in technologies (i.e., scientific testing and tracing methods) as well as audits allow the detection of a higher number of cases of food contamination faster. However, the recalls from organic food systems have increased at a greater scale relative to conventional food systems; suggesting that the increased ability of food systems to detect food contamination does not explain the increase in organic product recalls. This finding contradicts the results reported by Smith-Spangler et al. [10] who found no difference in risks between the conventional and organic food systems. These are certainly unexpected as well as alarming findings: organic products are highly trusted by consumers [48] as the demand has been steadily growing. Though our results should not be interpreted as a case against organic products, at the same time, it is troublesome that organic products pose potentially higher risks to consumers—an issue that needs to be addressed [49].

Secondly, organic products are more likely to be recalled for process reasons (i.e., bacterial contamination) rather than for design reasons (i.e., due to undeclared ingredients). The results support previous findings in the literature, for instance, a study by Strom [15] that highlighted that 87% of organic recalls since 2012 were for bacterial contamination, such as salmonella and listeria, rather than a problem associated with misleading labelling. In a similar vein, Pan et al. [50] asserted that "there was a twofold higher probability of Salmonella contamination in samples from growers or vendors who stated that they used organic farming practices compared with samples from those using conventional farming practices." The reason for increased levels of process errors might be explained in several ways [15,50]. Organic products have more constraints on how to handle contamination risks. For instance, organic farms are not allowed to use commercial fertilizers, and the use of manure increases the risk of *E. coli*. Second, organic production is also under more certification scrutiny. This increased scrutiny means that the producers are less likely to omit ingredients and allergens from their labels. At the same time, the increased scrutiny does not equally well address risks with process issues (such as bacterial contamination).

Third, processed food product recalls have a relatively higher propensity of design errors than process errors (high D/P ratio) compared to unprocessed food product recalls (this finding is applicable for both conventional and organic products). This means that processed products have a higher propensity of risk in the design (and are recalled for design reasons) rather than on the process aspects compared to the unprocessed products. For instance, unprocessed organic food has the lowest the D/P ratio, suggesting that most cases of recalls are caused by process aspects. As mentioned earlier, this is likely due to the specific practices of the organic production system (i.e., more constrains in handling contamination risks due to a reduced or forbidden use of pesticides that affects the effectiveness of weed management). This finding shows a clear contrast between processed conventional food products (dominated by design errors) and unprocessed organic food products (dominated by process errors). The result is insightful given the fact that organic food products are claimed to be tastier, healthier, safer and more nutritious than conventionally processed food products due the methods used to grow them (i.e., free from pesticides, fertilizers, or other unnatural substances). However, such reduced risk levels of potential chemical contamination are offset by increased risks related to other types of contamination during the production and supply chain processes.

## 5.1. Practical Implications

Our study provides several important strategic and operational implications for the management of organic food production systems.

At the policy level, we assert that while there exist well-established inspection, testing and certification systems such as the ISO 22000 standard for safety in food chains and several certification programs for organic produce [35], these schemes might be failing to prevent the risks of increasing contamination and consequently, recalls of organic products. Apart from the risk posed to consumers [51–53], the increasing recalls of organic products' recalls might also hamper the transition towards sustainable agricultural practices. Sustainable practices are an important part of the UN's SDG agenda, i.e., as SDG2 to end hunger assumes a rapid shift toward sustainable agriculture and sustainable practices that are inherent in organic production [54]. For these, and many other reasons, policy makers should reconsider the monitoring and control mechanisms and address the increasing level of organic product recalls by focusing on prevention measures that are specifically designed for organic food systems. There may be a greater need for technology to be leveraged in organic food production systems, to manage the complexity of these food systems. The investment and innovation of monitoring and control should also translate to innovations and the digitalization of inspections, monitoring, testing and auditing [30,55].

At the firm level, the findings imply that managers are ill-prepared to handle the complexities associated with organic food production systems. For supply chain managers, for instance, it means that higher levels of scrutiny of organic suppliers might be necessary and that supply chain partners in these supply chains might pose an increasingly higher risk. Managers should also be aware that whilst design risks are well managed, process risks are prevailing in organic food systems. In selecting supply chain partners, a high level of system traceability, for instance, is critical for reliable supply. The technological competence of firms also reduces the overall cost of product recalls. For example, the identification and traceability of contaminated batches is made easier with the use of a warehouse management system (WMS) or enterprise resource planning system (ERP), allowing the identification of batches/pallets of products.

There are substantial constraints on how organic food production systems can adapt and react to production challenges; the mechanisms will not necessarily be clear to managers from a conventional food production system background and may require greater planning and scrutiny. Further attention also needs to be applied to the management of recalls after they occur. The opportunity for cross-contamination (e.g., in a meat processing plant) and the contamination of unprocessed foods (e.g., such as lettuce or other produce) shows that traceability technologies still need further work. We also need to develop

methods of processing, handling and transporting organic foods that may limit potential costs; e.g., by improving traceability or reducing the batch sizes so that fewer discrete items may be affected by a food safety concern or recall. Insurance for food recall risks is scarce, often due to the extreme costs that would be incurred by a Class I recall event over a nation [44] and thus, resolving this issue might also reduce the costs of insurance cover for an organic food recall.

Any food product recall (including organic food) is a critical firm strategy that not only involves significant costs and loss of reputation, but also a food safety issue for consumers [51]. While consumers are better aware of the risk associated with organic food product recall [52] through social media nowadays, the elimination of such recall is unlikely to occur in the future. Firms are now adopting advanced technology such as whole-genome sequence to identify the exact cause of the recall [56]. It is also important to improve the efficacy of the food product recall system by better designing recall messages to consumers, alerting consumers about the potential hazards related to recall food products, and improving consumer response to recall products [56].

### 5.2. Limitations and Future Research

While our study used a large US dataset, representing a large and sophisticated market, it remains a single market. There are other markets (e.g., the European Union,) that often have a deeper and richer history of organic and alternative food production systems. It may be the case that there are other factors at play, such as a large and rapid shift towards organic food systems by firms who are unable to manage the complexities which must be managed in this environment. Our study also focused on determining the errors (recalls). Scrutinizing the sources of errors and where the errors emerge in the product life cycle, is outside of the scope of this study. The investigation has been limited to an analysis relating to the recalls and the production system only and has excluded a related analysis of, for instance, consumer awareness of and concern about errors in organic food production systems as well as excluding managerial decision making in product recall management (i.e., the initiation of voluntary versus mandatory recalls). Further research might address these limitations related to the scope of our study and a plethora of other questions related to the product safety of organic and conventional produce.

Further research is, for instance, needed to understand the transition from conventional towards organic produce. Recent trends suggests that the market is growing and also that producers are transitioning towards organic produce in expectation of future demand [57]. Future research might include these factors by controlling for the adoption and entry of new firms to the market, or with a cross-sectional study involving multiple regions with different histories and levels of industrial concentration (which may indicate the presence of new and small entrants) in both conventional and organic food production systems. Further research might show whether this is inherent or whether it may be affected by an increasing number of facilities adopting organic food production systems and finding themselves underprepared for the additional process challenges they experience in terms of greater complexity [50], process choices and higher labour challenges [38,40].

The use of technologies in food systems remains key for food safety [58]. Further studies are needed to understand the role of technologies for the rapid determination of sources of contamination and a rapid targeted recall of the contaminated products (rather than a large-scale recall). Targeted recalls also have an impact on food waste and future research can prioritize specific technologies and managerial practices that can prevent recalls in organic food systems. The role of consumers in product recalls also needs further scrutiny. It is timely to consider how consumers can become a more central part of food safety and food product recall systems, for instance through the use of home-based detection technologies for food contamination.

## 6. Conclusions

Drawing on a dataset of 2721 observations from 2010 to 2017 of US product recalls of conventional and organic food products, our paper addressed two food-related risks: design risk (a risk that is present at the development stage of the food product, such as the use of unapproved ingredients or the omission of some ingredients on the food label) and process risk (a risk within the supply chain, such as the contamination of food products with salmonella or *E. coli*). The paper demonstrates that organic products are becoming less safe and that organic products are recalled at higher rate. Organic products are also more prone to process risks rather than design risks. Our results indicate the challenges of transitioning to and managing organic food production systems. This paper also paves the way for further research in important areas—such studies that address producers, transitions from conventional to organic food systems; studies of the improvement and alignment of the use of technologies for product recalls and the role of consumers in participating in the detection of food risks in supply chains. Despite a somewhat negative light that these results put on organic products, the results should not be used to discourage consumers to shy away from organic produce. Rather, the results should be understood as a call for improved scrutiny and governance of organic food systems.

**Author Contributions:** Conceptualization, literature review, methodological approach and writing, M.C., P.C., D.P., X.Z. and L.C.W.; data collection, P.C. and X.Z.; data analysis, M.C., P.C. and D.P. All authors have read and agreed to the published version of the manuscript.

**Funding:** This research received no external funding.

**Institutional Review Board Statement:** Not applicable.

**Informed Consent Statement:** Not applicable.

**Data Availability Statement:** Publicly available datasets were analysed in this study. This data can be found here: http://www.fda.gov/Safety/Recalls/ArchiveRecalls/default.htm (accessed on 20 September 2019). The coding of the data is available upon request from corresponding authors.

**Conflicts of Interest:** All authors declare no conflict of interest.

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
