# Peer review of "Is Organic Food Becoming Less Safe? A Longitudinal Analysis of Conventional and Organic Product Recalls"

_sustainability, doi:10.3390/su132413540_

Round 1

Reviewer 1 Report

After reviewing the article, I believe that the research should be better constructed and grounded at a theoretical level; more current and solid research has been collected. 

Reviewer 2 Report

The paper is well structured: the research question are presented clearly as well as the results and discussions.

However, the reasoning behind an expressed implication does not emerge sharply, in particular it would be useful to clarify how the authors arrive at the claim that organic products have a higher proportion of recalls than non-organic products.

Reviewer 3 Report

The topic of the article is interesting, but it has many shortcomings.
1. First of all, the article lacks a chapter on research methods (there is only "Data collection" and "Data preparation and Coding")
2 . Line 281-289 should be moved to research methods. 
3. The data in the "Result" section are simple % data . There is a lack of statistical significance in the data presented. One should be tempted to do a deeper analysis of this data (e.g. according to Reasons for Recall).  You could also show the differences betweenProject Reasons and Process Reasons, using e.g. discriminant analysis. 
(4) Forecasting of future recalls of organic products could be done. 

Round 2

Reviewer 3 Report

The authors did not correct the article as directed, but provided clarifications regarding the defense of their version of the article.